# Evaluation of the Effectiveness of a Matrix of Exogenous Proteases in the Nutrition of Shrimp *Penaeus vannamei*

**DOI:** 10.3390/ani15101410

**Published:** 2025-05-13

**Authors:** Joice Teixeira Souza, Maria Érica da Silva Oliveira, Ana Elidarly da Cunha, Vanessa Maria Freitas da Silva, Ruan Arthur Nunes de Araújo, Mário Augusto Monteiro Silva, Raimundo Audei Henrique Júnior, Marcos Aurelio Victor de Assunção, Ana Cecília Araújo Lopes, Daniel Pigatto Monteiro, Thiago Pereira Ribeiro, Moacir Franco de Oliveira, Matheus Ramalho de Lima

**Affiliations:** 1Animal Science Department, Campus Mossoro, Federal Rural University of the Semi-Arid Region (UFERSA), Mossoro 59625-900, RN, Brazil; joice.souza@alunos.ufersa.edu.br (J.T.S.); erikinhaoliver10@yahoo.com.br (M.É.d.S.O.); ana.cunha45502@alunos.ufersa.edu.br (A.E.d.C.); vanessa.silva76996@alunos.ufersa.edu.br (V.M.F.d.S.); ruan.art0@gmail.com (R.A.N.d.A.); mario.silva58169@alunos.ufersa.edu.br (M.A.M.S.); henriquejunior9999@gmail.com (R.A.H.J.); marcos.assuncao@alunos.ufersa.edu.br (M.A.V.d.A.); ana.lopes81430@alunos.ufersa.edu.br (A.C.A.L.); moacir@ufersa.edu.br (M.F.d.O.); 2Animal Science Department, Campus Technological Innovation and Precision Animal Production Group (CNPq/UFERSA), Mossoro 59625-900, RN, Brazil; 3R&D in Nutrition at TECTRON, Toledo 85902-010, PR, Brazil; nutricao@tectron.com (D.P.M.); nutricao4@tectron.com (T.P.R.)

**Keywords:** shrimp, exogenous enzymes, protease, performance, fish quality, histology

## Abstract

In this study, we investigated the inclusion of a mixture of enzymes called proteases in shrimp diets with the aim of verifying whether the addition of these enzymes could increase feed efficiency and improve shrimp development, shrimp yield, and meat quality after slaughter compared to traditional diets. The experiment was carried out with four types of feed, two without enzyme supplementation and two with supplementation. Our results showed that the shrimp fed diets with enzymes grew faster and made better use of the feed, resulting in greater weight gain and meat yield after slaughter. Analysis of the intestine also revealed that these prawns had a better ability to absorb nutrients. In addition, the shrimp that consumed the diet with the proteases maintained the quality of the meat during storage. Therefore, we have concluded that the inclusion of these enzymes in diets can bring significant benefits to shrimp farming, such as greater productivity and better use of feed, contributing to more efficient and sustainable production.

## 1. Introduction

From a nutritional point of view, the protein sources used to feed aquatic organisms must meet important requirements, such as high protein content, adequate amino acid profile, high digestibility, good palatability, absence of anti-nutritional factors, as well as ensuring safe supply and affordable costs [1,2].

For these nutrients to be effectively utilized, it is essential to consider the functioning of the animal’s digestive system. In the case of shrimp, this system is made up of the stomach (foregut), hepatopancreas (midgut), and intestine (hindgut), which play a fundamental role in protein digestion, amino acid metabolism, and immunity [3].

In addition, an essential component in this process is digestive enzymes, which are important in various stages of the shrimp’s physiological processes, especially in immunity and development [4]. Besides helping digestion, endogenous enzymes promote the breakdown of proteins and lipids into simpler units, facilitating the absorption of nutrients.

Several proteases, such as trypsin, chymotrypsin, leucine aminopeptidase, and cathepsin L, are involved in protein digestion. In shrimp, digestion takes place in different parts of the digestive system and is essential for protein degradation. These enzymes show optimum activity in specific pH and temperature ranges [5].

Although endogenous enzymes play a fundamental role in protein digestion, the efficiency of this process may be limited by factors such as the animal’s feeding behavior, habitat, and others [6]. Therefore, the supplementation of exogenous enzymes in shrimp diets has proven to be an effective strategy to aid in nutrient digestion and utilization, increase bioavailability, and eliminate antinutritional factors present in the diets [7].

The supplementation of exogenous protease in diets for different species has shown significant benefits in digestibility and zootechnical performance. In broiler chickens, the study by Yi et al. [8] evaluated the effects of an alkaline protease derived from *Bacillus licheniformis*, supplemented at different levels (0, 100, 200, 300, and 400 mg/kg). The results showed that protease supplementation significantly improved growth performance by reducing the feed conversion ratio and increasing final body weight. Additional benefits were observed in meat quality, including higher breast yield and lower drip loss, as well as enhanced protein digestibility and increased trypsin activity in the jejunum.

Studies on subtilisin protease derived from *Bacillus licheniformis* also indicate improvements in feed conversion and productive efficiency [9]. Beyond poultry, the addition of proteases in swine diets has been widely studied, demonstrating gains in growth and nutrient absorption [10]. Research on enzymes in animal nutrition continues to evolve, aiming to understand their interaction with intestinal microbiota and their potential to reduce antibiotic use by improving gut health and decreasing disease incidence [7].

In this context, the search for more efficient enzymatic strategies has led to the development of protease blends, which combine different types of enzymes to optimize protein digestion in various species. Among these strategies, protease blends, which combine acid, neutral, and alkaline proteases acting at different pH ranges, stand out for offering greater digestive efficiency throughout the gastrointestinal tract of fish and crustaceans [6]. The interaction between alkaline and neutral proteases in the digestive tract may generate a synergistic effect, increasing the free amino nitrogen (FAN) content compared to the isolated supplementation of a single protease [11]. This enhances protein digestion and nutrient absorption, benefiting the animals’ zootechnical performance.

The supplementation of exogenous proteases in shrimp diets has been shown to be effective in several aspects, such as increasing protein digestibility, protein synthesis, and the reduction of nitrogen compounds. These benefits result in significant improvements in zootechnical performance, including greater weight gain and productivity, as evidenced by several studies [12,13,14,15]. In addition, supplementation stimulates the release of endogenous digestive enzymes, optimizing protein solubilization and minimizing undesirable fermentation in the shrimp’s hind gut [16].

Recent studies have shown that exogenous proteases not only improve zootechnical parameters, such as weight gain and feed conversion, but also contribute to intestinal health and meat quality [17,18,19,20,21].

The use of exogenous enzymes in plant-based protein diets has shown benefits in inactivating anti-nutritional factors (ANFs), improving the rate of absorption and digestion of nutrients in the diet [17,21,22,23]. Similarly, in diets with reduced fishmeal, the results are also positive, reflecting cost savings and better results in zootechnical performance, which translates into greater growth and survival of the animals [22,24,25,26].

By improving protein digestion and the absorption of essential amino acids, supplementation with a blend of exogenous proteases can not only optimize shrimp performance but also allow the use of diets with more sustainable and economical ingredients, without compromising the quality of the product.

In this context, this study was designed to evaluate the effects of supplementing a blend of acid and alkaline proteases in the diet of *Penaeus vannamei* shrimp, considering performance, yield, meat quality, and intestinal histology data.

## 2. Materials and Methods

### 2.1. Animals and Experimental Units

The post-larvae of *Penaeus vannamei* were acquired from a farm located in Aracati-CE, Brazil, and transported to the Aquaculture Sector of the Federal Rural University of the Semi-Arid-UFERSA, in Mossoro-RN, Brazil, where the experiment was conducted (CEUA protocol 33/2022).

The experimental units (EUs) consisted of PVC-coated multifilament nylon net tanks with a 5 mm mesh and a useful volume of 1 m^3^, equipped with an aeration system and individual feeding trays. The net tanks were installed in masonry tanks with an area of 15 m^2^ and a depth of 1.2 m.

After the nursery period, the shrimps were weighed (0.5 g ± 0.02) and randomly assigned to the treatments. The experiment consisted of 8 masonry tanks, corresponding to 32 EUs (net tanks), divided into 4 treatments with 8 repetitions. The EUs were stocked with 80 animals/EU, totaling 640 animals per treatment, resulting in a stocking density of 80 animals/m^2^.

The water used in the tanks was collected from an artesian well, and partial water changes were carried out routinely. Water quality parameters were measured twice a week.

### 2.2. Experimental Design, Diets, and Management

The experiment was organized in a completely randomized design, using a 2 × 2 factorial model, with 4 treatments. The treatments consisted of four diets: Positive Control (PC), based on the nutritional recommendations for the animals evaluated according to the recommendations of the National Research Council [27]; and Negative Control (NC), which presented a nutritional reduction based on 100% of the blend of acid and alkaline proteases (energy, crude protein, methionine + cystine, methionine, lysine, and threonine). These diets were combined with 250 g/t of the acid and alkaline protease blend, resulting in the following treatments: PCE (PC + 250 g/t of Enzymes) and NCE (NC + 250 g/t of Enzymes).

The diet of the Negative Control (NC) group was formulated with a reduced inclusion of fishmeal to evaluate the ability of enzymatic supplementation to compensate for the lower availability of animal protein. The selection of fishmeal as the target for nutritional reduction was based on its high digestibility and relevance as a protein source in aquafeeds, allowing for the assessment of whether the blend of acid and alkaline proteases could enhance nutrient digestibility and maintain the animals’ zootechnical performance, even with a formulation containing lower levels of this strategic ingredient.

The proteases used in the present experiment were supplied by Tectron, where the acid protease was from *Aspergillus niger*, and the alkaline one from *Bacillus subtilis*. The enzymatic activity of these proteases is 20,000 U/g.

The feed and nutritional composition are shown in Table 1. The feed produced was extruded, and the 1.6 mm pellets were dried in an air circulation system, with the dust removed in a vibrating station, after which the feed produced was stored as recommended.

The shrimps were fed 6–8% of their body weight twice a day (at 7 a.m. and 3 p.m.) for 90 days. Every 15 days, the animals were captured in 20% of their initial biomass using a dip net. After capture, the shrimp were carefully handled to minimize stress and transferred to a tray for weighing. Based on the collected biometric data, the feeding rate was adjusted according to the average weight gain per treatment, following the nutritional recommendations for the species.

The reduction in fishmeal content from 9.788% in the positive control (PC) diet to 9.528% in the negative control (NC) diet was based on the matrix value provided by the enzyme manufacturer. According to the manufacturer, the protease blend contributes approximately 1% additional protein, in addition to improvements in the availability of metabolizable energy and key amino acids such as methionine, lysine, threonine, and methionine + cystine. To reflect this added nutritional contribution, fishmeal was proportionally reduced, and the formulation was rebalanced using soybean meal and corn to maintain nutritional profiles across all treatments. This precision formulation approach aimed to evaluate whether the enzyme could compensate for small reductions in high-quality animal protein sources under realistic feed formulation conditions.

### 2.3. Zootechnical Performance

After 90 days, the animals were harvested, weighed, and measured for performance calculations according to the following parameters and formulas: initial weight: IBW (g), live weight: BW (g), feed consumption: FI (g), weight gain: BGW (g) = (BWt − BWi), feed conversion: SFR (g/g) = (feed consumption/BWG), and final survival: Survival rate (%). Wi and Wt indicate the initial and final weight of the shrimp.

Feed consumption was obtained from the difference between what was offered and the leftovers per experimental unit. For feed conversion, weight gain, and feed consumption per experimental unit were considered.

### 2.4. Yield of Headless and Shell-Less Shrimp

The animals were slaughtered by heat shock, and after the weighing and counting procedures, 30 shrimps per treatment were measured and weighed individually. The head and shell-less were removed to obtain the yields of headless shrimp, YieldSC (%) = weight of headless shrimp/weight of whole shrimp × 100, and headless and shell-less, YieldSSC (%) = weight of headless and shell-less/weight of whole shrimp × 100.

### 2.5. Analysis of Centesimal Composition

For the centesimal composition of the shrimp, the biological samples were previously dried at 55 °C for 72 h in an oven with forced ventilation. After drying, these samples were milled in a hammer mill, and then the analyses of moisture, ether extract by the Soxhlet method [28], crude protein (N × 6.25) by the Kjeldahl method, and mineral matter were carried out, following the prescriptions of Ref. [29].

### 2.6. Meat Quality Analysis

The analyses were carried out on the headless and shell-less shrimp while they were still fresh. The shrimps were then frozen using the slow freezing method, glazed, frozen again using the slow freezing method, packed in plastic packaging, separated by treatment, and kept at −18 ± 2 °C for quality assessment for 210 days.

For the drip loss analysis, the frozen samples were thawed under refrigeration at 4 °C for 24 h, weighed, and calculated according to Gonçalves et al. [30].

The water retention capacity (WRC) was determined according to the methodology described by Hamm [31]. Weight loss by cooking (WL), according to Osório et al. [32], with minor adjustments, the samples were cooked on a grill until they reached an internal temperature of around 75 °C. The result was given as a percentage of water loss during the thermal process. The Shear Force (SF) was measured using a TEXTURE ANALYZER TA-XT-125 (Stable Micro Systems, Godalming, UK), coupled to the Warner-Bratzler device, which expresses the force in kgf/cm^2^ [31].

### 2.7. Histological Analysis

For the histological study of the intestine, four shrimps were collected per treatment and fixed in a 10% buffered formaldehyde solution for 24 h at room temperature. After fixation, the samples were dehydrated in an increasing series of ethyl alcohol, clarified with xylene, and embedded in paraffin [33]. Serial cross-sections (5 µm thick) were obtained using a Leica RM 2125 RT^®^ (Leica Microsystems, Wetzlar, Germany) microtome, mounted on glass slides, and stained with hematoxylin and eosin (H&E) [34]. The slides were analyzed under a light microscope (LEICA DM 500 HD, Leica Microsystems, Wetzlar, Germany) equipped with a digital camera (LEICA ICC50W, Leica Microsystems, Wetzlar, Germany), and images were captured using the LAS EZ Ink program (version 3.4.0 (2025)).

A single section of the mid-intestine was analyzed per shrimp. Intestinal wall thickness (IWT) and villus height (VH) were measured using ImageJ 1.54 software (National Institutes of Health, Bethesda, MD, USA). For villus height measurements, 10 villi were assessed per shrimp, with a total of four shrimps per treatment. The AV/EPI ratio was used to determine the intestinal absorption surface [35].

### 2.8. Statistical Analysis

The data obtained were analyzed using R software (Version 2024.04.2+764) and compared using analysis of variance (ANOVA) in a factorial experimental design, followed by the Tukey test. When significant differences were identified (*p* < 0.05), the *t*-test was used for further comparisons.

## 3. Results

All experimental groups showed comparable survival rates, approximately 79%, with no statistically significant variation (*p* > 0.05) between treatments (Table 2).

The metrics, including body weight (BW), body weight gain (BWG), daily growth rate (SGDAY), weekly growth rate (SGWEEK), feed efficiency (FE), and feed conversion ratio (SFR), were all significantly affected (*p* < 0.05) by the incorporation of protease into the dietary formulations, under both PCE and NCE conditions.

Shrimps that received protease-enriched diets showed statistically significant improvements in BGW, SGDay, and SGWeek (*p* < 0.05) compared to those that consumed diets without protease supplementation.

The feed conversion ratio (SFR) of shrimp fed PCE and NCE diets showed an increase (*p* < 0.05) compared to shrimps that were fed PC and CN diets.

Table 3 shows that the diets supplemented with the protease matrix resulted in a higher production of shrimps per unit area, as well as a higher yield of headless and shell-less shrimp (*p* < 0.001). The protease blend matrix provided more favorable results in terms of yield for the shrimp fed the NCE and PCE diets, with emphasis on the NCE group.

The total body composition of *P. vannamei* fed the experimental diets is shown in Table 3. In general, only the lipid content suffered significant interference (*p* < 0.05) from the diets in the different treatments.

Table 4 shows the results obtained from the quality analyses of frozen shrimp stored at −18 °C for 210 days. There was a significant difference (*p* < 0.05) between treatments only in moisture content and WRC during the storage period. The results suggest a higher water retention capacity in the PCE and NCE treatments (*p* = 0.0266). Cooking weight loss (WCL) was lower in the PCE and NC treatments, with a significant difference (*p* > 0.05) between the treatments at the end of the experiment.

With regard to drip loss, it can be seen (Figure 1) that there was no significant difference (*p* > 0.05) between the treatments during the storage of frozen shrimp, but lower averages were found in the PC (11.07%) and PCE (16.68%) treatments.

From the photomicrographs of the intestinal histology of *P. vannamei* shown in Figure 2, it is evident that the morphology of the intestine was affected by the experimental diets, particularly in the NC (C) and NCE (D) groups.

The morphometric measurements of the intestine are shown in Figure 3. Significant differences were found in villus height (*p* = 0.021), intestinal wall thickness (*p* = 0.002), and absorption area (*p* = 0.007) in the PCE, NC, and NCE groups. No villi were found in the PC group.

An unexpected outcome was the absence of discernible intestinal villi in shrimps from the PC group. Histological examination revealed indications of epithelial erosion and detachment in the mucosa, which hindered the visualization and quantification of villus height and absorptive surface area. In contrast, the underlying submucosa and muscularis layers appeared intact. This mucosal damage may have arisen from localized inflammation, inadequate sample handling, or degradation during fixation. Although this phenomenon was not observed in the other treatment groups, it restricted our ability to assess morphometric parameters in the PC group.

## 4. Discussion

One of the functions of protease in the diet of aquatic organisms is the enzymatic hydrolysis of proteins into individual amino acids and peptides, as well as acting in the degradation of anti-nutrients, improving protein digestion, compensating for the deficiency of endogenous enzymes, especially in young animals, and reducing nitrogen excretion [20,36,37].

Exogenous proteases can compensate for the deficiency of endogenous enzymes by breaking down large molecules that are difficult to digest. The breakdown of these macromolecules aids in the formation of bioactive peptides and the degradation of anti-nutrients. These effects positively influence zootechnical performance and animal health, as well as generating economic gains in production [20,38].

The supplementation of protease, either individually or in combination, in low fishmeal diets has been shown to enhance growth performance, improve nutrient utilization, and increase weight gain in Pacific white shrimp, indicating the potential for reducing fishmeal in feed [12,39].

This trend is further supported by findings from different experimental designs. Yao et al. [39] formulated nine diets to evaluate the effects of functional additives on Pacific white shrimp. A positive control diet (20% fish meal) and a negative control diet (10% fish meal) were used as references. The negative control diet was supplemented with a protease complex (175 mg/kg), a multi-carbohydrase (100 mg/kg), and a blend of microencapsulated organic acids (825 mg/kg), either individually or in combinations. The results demonstrated that the individual or combined supplementation of these additives, particularly protease, improved the growth performance and nutrient utilization of shrimp fed low fish meal diets.

The study by Li et al. [12] evaluated three dietary treatments: a high fish meal diet, a low fish meal diet, and a low fish meal diet supplemented with 175 mg/kg of protease. The results demonstrated that protease supplementation in low fish meal diets enhanced shrimp growth performance, suggesting the potential to reduce fish meal inclusion in aquafeeds.

On the other hand, in a recent study, Coelho et al. [19] reported that protease supplementation had no significant effect on the performance, digestibility, and nutrient retention in juvenile *Penaeus vannamei* because, under the experimental conditions, the addition of this enzyme did not yield measurable improvements in growth, feed conversion, or nutrient assimilation. The study evaluated exogenous enzyme supplementation in fish meal-free and phosphorus-limited diets through two feeding trials. While phytase supplementation showed significant benefits in growth performance and phosphorus digestibility, protease did not exhibit similar positive effects.

However, another positive aspect observed in our study was the maintenance of shrimp survival, an important factor for the viability and profitability of shrimp production. The absence of an effect of protease supplementation on final batch survival suggests that the addition of 250 g/t of protease did not cause adverse effects on shrimp survival.

In addition to survival, yield is a key factor in the profitability of the production chain. The headless and shell-less shrimp, marketed as “shrimp fillet”, is the most economically valuable product; thus, the headless shrimp yield is an important measure to evaluate the proportion of usable meat after processing the shrimp.

In this study, the supplementation of exogenous protease resulted in a significantly higher headless shrimp yield compared to non-supplemented diets. In addition, headless and shell-less yield also benefited from protease supplementation, showing values of 53.03% and 52.21% in the NCE and PCE groups and 51.36% and 50.27% in the NC and PC groups. It is important to note that gains in headless and shell-less yield also benefited from protease supplementation, which has positive economic implications for the production chain.

The effectiveness of exogenous protease in improving yield headless and shell-less, also benefited from protease supplementation, was particularly evident in the NCE diet, indicating a positive interaction between diet and supplementation. Considering that headless and shell-less processed shrimps are widely marketed, finding strategies that promote an increase in the yield of the final product represents significant economic gains.

Freezing is the most commonly used method to inhibit microbial growth, slow biochemical reactions, and extend the shelf life of shrimps. In addition to freezing, pH is a crucial parameter for assessing the quality of fish products, as it directly affects protein stability, is strongly related to fish deterioration, and significantly influences the water-holding capacity (WHC) of the meat [40]. Lower pH levels reduce WHC due to the charge shielding effect of lactic acid on proteins, leading to drier and less tender meat [41].

The highest average pH found in this study was 6.67 in the PC group. According to Brazilian legislation, this product would be fit for consumption, since the ideal pH for shrimp meat, as indicated by this legislation, should be below 7.85 [42].

In turn, Chouljenko et al. [43] indicated that a pH of up to 7.95 classifies shrimp as fit for consumption.

Another important aspect to assess is moisture. Fish is, by nature, a food with a high moisture content, and maintaining this parameter is related to yield, nutritional composition, and overall appearance [44]. In this sense, better moisture results were presented in the NCE group, which maintained linearity in moisture content throughout the storage period, demonstrating that there were fewer losses related to dehydration, yield, and nutrients.

Although high moisture content is associated with quality attributes, providing a more delicate flavor to the fish, this attribute can easily promote higher drip losses during storage.

Loss due to dripping affects juiciness, which is directly related to texture and flavor, as well as compromising the nutritional composition and generally being associated with an unpleasant appearance and texture of the fish [45]. In this study, there was no interference from the experimental diets in this parameter (*p* = 0.1487); however, lower losses were recorded in the PC and PCE diets. This result suggests that the experimental diets did not affect the weight of the shrimp after thawing and, consequently, the yield.

After processing, cooking weight loss (CWL) is a variable directly related to meat yield [46]. Samples of shrimp fed the NC, PCE, and NCE diets showed lower WCL values. These results corroborate the findings regarding meat yield in this study.

Samples of shrimp fed the NCE, PCE, and PC diets were found to have better water retention capacity (WRC) in the meat. WRC is one of the attributes that can influence the consumer’s purchasing decision, as it is related to the texture of the meat. As one of the most important quality parameters in shrimp processing, food with a good WRC tends to suffer lower losses during storage [47,48]. Mechanisms that help with WRC and prevent losses during storage and after cooking are extremely necessary to guarantee a good yield and the quality of the shrimp after processing.

Among the treatments, meat firmness (SF) did not vary considerably (*p* = 0.5893), showing that protease did not interfere with this parameter. Some authors, when studying different freezing techniques on samples of *P. vannamei*, also found no significantly different values for SF [46,48].

In the shrimp gut, supplemented exogenous enzymes can be an important nutritional element to optimize nutrient utilization, promote growth, and improve bowel functions [15].

The results obtained in the zootechnical performance and meat quality can be influenced by the effects of exogenous protease in the intestine of the animals, since the diets ingested by the organisms can affect the morphology and structure of the intestinal villi [49]. In this sense, the greater the length and width of the intestinal villi, the greater the surface of nutrient absorption. Therefore, increasing villi height and intestinal wall thickness can promote better digestion of feed by increasing the area of nutrient absorption in the intestine, showing a direct relationship with the specific growth of animals [50].

In the present study, the PCE group had a higher intestinal wall thickness (IWT) and lower villus height (VH) and, therefore, a smaller absorption area (AA), resulting in a smaller absorption area compared to the NC and NEC groups, which had higher AA. These results may indicate that the experimental diets positively influenced intestinal morphology. These findings corroborate the specific growth rate and feed efficiency observed in the NCE group, which showed higher weekly weight gain (SGWeek) and feed efficiency (FE), suggesting that the presence of a larger absorption area can improve feed digestion, optimize nutrient absorption, and contribute to animal growth, as described by Wei et al. [51].

Ensuring the efficacy of exogenous enzymes throughout the entire digestive tract of *Penaeus vannamei* necessitates a comprehensive understanding of the pH variability along the intestine. Previous research has substantiated the beneficial effects of exogenous protease supplementation—particularly in diets characterized by low fish meal content—on growth performance, nutrient digestibility, immune function, and gut morphology [18]. However, most of these studies predominantly focus on single types of proteases, which are often restricted to specific pH ranges. The present investigation advances this field by illustrating the advantages of combined supplementation of acid and alkaline proteases, thereby facilitating enzymatic activity across a broader pH spectrum. This dual-enzyme strategy optimizes protein hydrolysis throughout the digestive tract, enhancing nutrient absorption consistency and contributing to improved hematological parameters. Although prior studies have demonstrated the efficacy of protease mixtures in conjunction with other enzymes in augmenting feed efficiency and survival [52] and have reported that alkaline proteases derived from microbial sources can modulate gut microbiota and enhance the nutritional quality of feeds [53], our findings represent the inaugural evidence of the functional complementarity between acid and alkaline proteases in shrimp nutrition. Furthermore, the in vivo validation of enzyme efficacy presented here addresses the limitations associated with in vitro assessments, which do not consistently correlate with improved performance in live shrimps [54]. Consequently, this study introduces an innovative and practical framework for enzyme supplementation in shrimp diets, underscoring the potential of tailored enzyme blends as instruments for optimizing feed utilization, reducing costs, and promoting health within sustainable aquaculture systems [14].

From a practical perspective, it is essential to examine the economic implications of protease supplementation in shrimp diets. While the incorporation of exogenous enzymes incurs a modest increase in feed costs, the observed enhancements in growth performance, feed conversion ratios, and hematological health indicate that these expenditures are warranted. Notably, in diets with reduced fish meal content, the improved protein utilization afforded by the synergistic application of acid and alkaline proteases may offset the cost increase by diminishing the necessity for high-cost protein ingredients and enhancing overall feed efficiency. These findings substantiate the economic viability of employing tailored enzyme blends as a cost-effective strategy for improving productivity within shrimp farming systems.

## 5. Conclusions

This study investigated the effects of supplementing a blend of acid and alkaline proteases at a concentration of 250 g/t in the diets of *Penaeus vannamei*. The inclusion of enzyme supplementation significantly improved nutrient utilization, resulting in enhanced growth performance, increased yield, and improved meat quality. These findings underscore the efficacy of employing protease blends to optimize digestive efficiency and production metrics in shrimp aquaculture. Considering the growing demand for sustainable and efficient aquaculture practices, the application of proteases emerges as a promising nutritional strategy for enhancing productivity in intensive shrimp culture systems.

## Figures and Tables

**Figure 1 animals-15-01410-f001:**
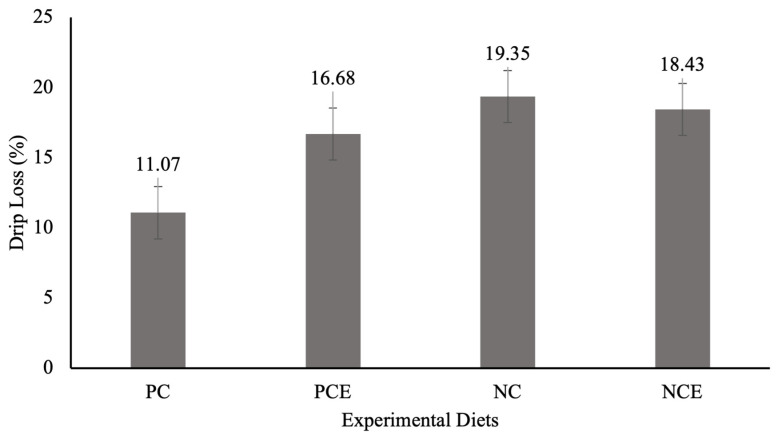
Effect of experimental diets on drip loss of *P. vannamei* shrimp during frozen storage at −18 °C. Results are presented as mean and standard error. Nutrition *p* = 0.0354. Enzyme *p* = 0.2894. N × E = 0.1487. C.V (%) = 25.82.

**Figure 2 animals-15-01410-f002:**
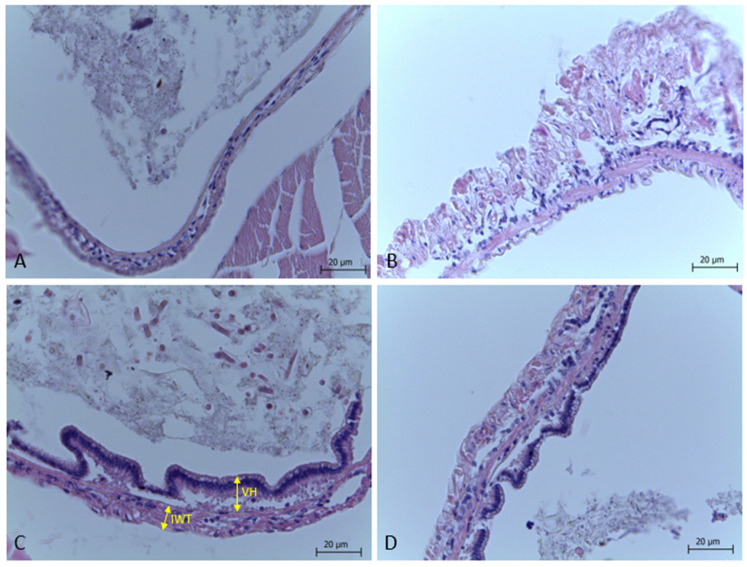
Photomicrographs of cross-sections of the intestines of *P. vannamei* stained with HE fed diets containing exogenous protease blend PCE and NCE (**B**,**D**) and diets without exogenous protease supplementation PC and NC (**A**,**C**). IWT: intestinal wall thickness; VH: villus height.

**Figure 3 animals-15-01410-f003:**
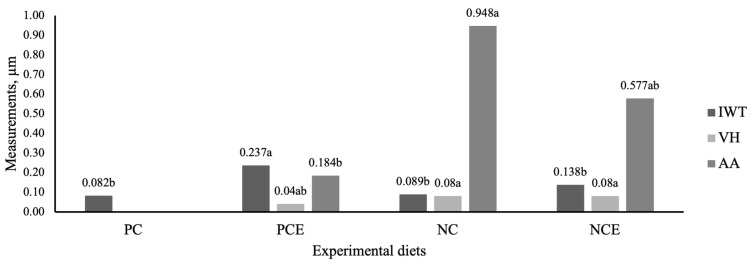
Effects of different diets on the wall thickness, villus height, and absorption area of the intestine of *P. vannamei*. Results presented are mean ± standard deviation (n = 5). Different letters indicate a significant difference (Tukey’s test, *p* ˂ 0.05). IWT: intestinal wall thickness; VH: villus height; AA: absorption area.

**Table 1 animals-15-01410-t001:** Composition and nutrient content of experimental diets.

Description	PC	PCE	NC	NCE
Soybean meal 45%	41.446	41.446	41.604	41.604
Corn 8%	31.421	31.421	32.080	32.080
Fishmeal 53%	9.788	9.788	9.528	9.528
Fish oil	3.500	3.500	3.074	3.074
Artemia biomass	4.075	4.075	4.075	4.075
Common salt	1.251	1.251	1.251	1.251
Premix	1.000	1.000	1.000	1.000
Vitamin C	0.600	0.600	0.600	0.600
L-Lysine HCl	0.380	0.380	0.255	0.255
L-Threonine	0.288	0.288	0.284	0.284
DL-Methionine	0.258	0.258	0.255	0.255
Seaweed	2.000	2.000	2.000	2.000
Soy lecithin	1.500	1.500	1.500	1.500
Potassium	1.200	1.200	1.200	1.200
Lactic acid	0.750	0.750	0.750	0.750
Sodium lactate	0.325	0.325	0.325	0.325
Adsorbent	0.100	0.100	0.100	0.100
Binder	0.035	0.035	0.035	0.035
Antioxidant	0.030	0.030	0.030	0.030
Antifungal	0.030	0.030	0.030	0.030
Inert	0.025	0.000	0.025	0.000
Protease blend	0.000	0.025	0.000	0.025
Total	100.000	100.000	100.000	100.000
**Calculated chemical composition**				
Moisture (%)	10.237		10.237	
Dry matter (%)	83.763		83.763	
Ashes (%)	7.513		7.513	
TDN (%)	56.208		56.208	
Gross energy (kcal/kg)	4200.000		4162.500	
Crude protein (%)	35.000		34.000	
Crude fiber (%)	4.768		4.768	
Acid detergent fiber (%)	6.983		6.983	
Neutral detergent fiber (%)	10.499		10.499	
Methionine (%)	0.800		0.775	
Met + Cys (%)	1.263		1.225	
Lysine (%)	2.400		2.335	
Threonine (%)	1.600		1.553	
Tryptophan (%)	0.426		0.426	
Arginine (%)	2.369		2.369	
Isoleucine (%)	1.473		1.473	
Valine (%)	1.610		1.610	
Aspartic acid (%)	3.709		3.709	
Glutamic acid (%)	5.721		5.721	
Grease (%)	5.676		5.676	
Linoleic acid (%)	0.684		0.684	
Linolenic acid (%)	0.097		0.097	
Arachidonic acid (%)	0.688		0.688	
Calcium (%)	0.760		0.760	
Phosphorus (%)	0.733		0.733	
Sodium (%)	0.600		0.600	
Potassium (%)	1.316		1.316	
Chlorine (%)	1.002		1.002	

PC: positive control; PCE: positive control + protease; NC: negative control; NCE: negative control + enzyme; PC: crude protein.

**Table 2 animals-15-01410-t002:** Starting weight (IBW, g/shrimp), live weight (BW, g/shrimp), feed consumption (FI, g/shrimp), weight gain (BGW, g/shrimp), specific growth rate per day (SGDay, %) and per week (SGWeek, %), food efficiency (FE, %), feed conversation ratio (FCR, g/g), and survival (Survival rate %) of shrimp fed diets without and with reduction (PC and NC) supplemented with 250 g/t of protease for 90 days.

Main Effects	IBW (g)	BW (g)	FI (g)	BWG (g)	SGDAY (%)	SGWEEK (%)	FE (%)	FCR (g/g)	Survival (%)
PC	0.57 ± 0.01	13.2 ± 1.33	25.8 ± 0.52	12.6 ± 1.33	15.0 ± 1.52	104.9 ± 11.1	48.9 ± 5.57	2.07 ± 0.22	78.1 ± 4.15
NC	0.56 ± 0.02	12.9 ± 1.76	25.5 ± 0.70	12.4 ± 1.76	14.7 ± 2.09	103.0 ± 14.7	48.4 ± 6.82	2.09 ± 0.28	78.9 ± 7.08
0 g/t protease	0.56 ± 0.02	11.9 ^b^ ± 0.57	25.8 ± 0.56	11.3 ^b^ ± 0.57	13.5 ^b^ ± 0.69	94.6 ^b^ ± 4.82	44.0 ^b^ ± 2.06	2.28 ^a^ ± 0.11	77.± 7.30
250 g/t protease	0.57 ± 0.02	14.5 ^a^ ± 1.00	25.5 ± 0.67	13.9 ^a^ ± 1.01	16.6 ^a^ ± 1.202	116.0 ^a^ ± 8.41	54.7 ^a^ ± 3.66	1.84 ^b^ ± 0.12	80.6 ± 3.41
Interaction									
PC	0.57 ± 0.02	12.1 ^b^ ± 0.63	26.0 ± 0.53	11.6 ^b^ ± 0.64	13.8 ^b^ ± 0.76	96.4 ^b^ ± 5.33	44.5 ^b^ ± 2.15	2.25 ^A^ ± 0.11	76.9 ± 3.95
PCE	0.57 ± 0.02	14.2 ^a^ ± 1.01	25.6 ± 0.43	13.6 ^a^ ± 1.02	16.2 ^a^ ± 1.21	113.4 ^a^ ± 8.52	53.3 ^a^ ± 4.23	1.89 ^b^ ± 0.15	79.4 ± 4.22
NC	0.56 ± 0.02	11.7 ^B^ ± 0.43	25.5 ± 0.52	11.1 ^B^ ± 0.44	13.2 ^B^ ± 0.52	92.7 ^B^ ± 3.66	43.6 ^B^ ± 1.99	2.30 ^A^ ± 0.10	78.4 ± 9.86
NCE	0.57 ± 0.02	14.8 ^A^ ± 0.94	25.4 ± 0.87	14.2 ^A^ ± 0.95	16.9 ^A^ ± 1.13	118.6 ^A^ ± 7.96	56.1 ^A^ ± 2.52	1.79 ^B^ ± 0.08	81.9 ± 1.89
Nutrition	0.30	0.78	0.16	0.76	0.76	0.76	0.37	0.52	0.33
Protease	0.54	<0.001	0.17	<0.001	<0.001	<0.001	<0.001	<0.001	0.16
Interaction	0.54	0.04	0.05	0.03	0.03	0.03	0.02	0.01	0.82
C.V. (%)	3.630	6.01	2.41	6.34	6.34	6.34	5.82	5.55	7.32

Results presented as mean ± standard deviation. Different superscript letters indicate significant differences among treatments (*p* < 0.05, ANOVA followed by Tukey’s test). In the main effects: a, b differ in relation to nutrition (PC vs. NC) and protease (0 g/t vs. 250 g/t). In the interaction: a, b (PC vs. PCE); A, B (NC vs. NCE). C.V. (%) coefficient of variation.

**Table 3 animals-15-01410-t003:** Yield of headless shrimp (YieldSC, %), yield of shrimp headless and shell-less (YieldSCC, %), and body composition of *P. vannamei* fed diets without and with nutritional reduction (PC and NC) supplemented with 250 g/t of protease over 90 days of cultivation.

Main Effects	YieldSC (%)	YieldSCC (%)	Moisture (%)	Protein (%)	Lipids (%)	Ash (%)
PC	62.2 ± 2.25	51.2 ^b^ ± 1.56	75.9 ± 2.20	19.3 ± 1.47	0.26 ± 0.03	4.5 ± 0.41
NC	62.4 ± 2.23	52.2 ^a^ ± 1.51	76.4 ± 1.13	18.6 ± 1.46	0.27 ± 0.05	4.5 ± 0.22
0 g/t protease	60.5 ^b^ ± 0.91	50.8 ^b^ ± 1.37	75.9 ± 2.29	19.1 ± 1.82	0.28 ± 0.04	4.4 ± 0.37
250 g/t protease	64.0 ^a^ ± 1.73	52.6 ^a^ ± 1.29	76.4 ± 0.92	18.7 ± 1.07	0.25 ± 0.04	4.6 ± 0.26
Interaction						
PC	60.5 ^b^ ± 0.83	50.3 ^bY^ ± 1.32	74.9 ± 2.97	20.1 ± 1.80	0.25 ± 0.03	4.2 ± 0.43
PCE	63.9 ^a^ ± 1.96	52.2 ^ay^ ± 1.12	76.8 ± 0.83	18.4 ± 0.22	0.27 ± 0.03	4.8 ± 0.10
NC	60.5 ^B^ ± 1.00	51.4 ^BX^ ± 1.20	76.8 ± 1.36	18.2 ± 1.51	0.31 ± 0.04	4.7 ± 0.16
NCE	64.2 ^A^ ± 1.49	53.0 ^Ax^ ± 1.33	76.0 ± 0.97	19.0 ± 1.50	0.23 ± 0.03	4.4 ± 0.21
Nutrition	0.481	0.00	0.62	0.43	0.73	0.90
Protease	<0.001	<0.001	0.60	0.62	0.14	0.37
Interaction	0.52	0.56	0.23	0.17	0.03	0.02
C.V. (%)	2.23	2.41	2.31	7.53	12.0	5.69

Results presented as mean ± standard deviation. Different superscript letters indicate significant differences among treatments (*p* < 0.05, ANOVA followed by Tukey’s test). In the main effects: a, b differ in relation to nutrition (PC vs. NC) and protease (0 g/t vs. 250 g/t). In the interaction: a, b (PC vs. PCE); A, B (NC vs. NCE); X, Y (PC vs. NC), x, y (PCE vs. NCE). C.V. (%) coefficient of variation.

**Table 4 animals-15-01410-t004:** Effects of experimental diets on pH, moisture, protein, water retention capacity (WRC), cooking weight loss (WCL), and shear force (SF) of *P. vannamei* shrimp meat stored frozen at −18 °C for 210 days.

Main Effects	pH	Moisture (%)	Protein (%)	WRC (%)	WCL (%)	SF (kgf)
PC	6.57 ^a^ ± 0.37	76.1 ^a^ ± 1.29	23.4 ^a^ ± 3.05	70.78 ^a^ ± 3.69	60.1 ^a^ ± 2.23	2.05 ^a^ ± 0.94
NC	6.42 ^a^ ± 0.24	76.7 ^a^ ± 1.23	22.5 ^a^ ± 2.47	72.5 ^a^ ± 3.80	58.9 ^a^ ± 2.01	2.11 ^a^ ± 0.88
0 g/t protease	6.54 ^A^ ± 0.35	76.4 ^A^ ± 1.11	22.8 ^A^ ± 2.65	70.2 ^B^ ± 4.26	59.8 ^A^ ± 2.00	2.05 ^A^ ± 0.91
250 g/t protease	6.45 ^A^ ± 0.25	76.4 ^A^ ± 1.42	23.1 ^A^ ± 2.86	73.1 ^A^ ± 3.57	59.2 ^A^ ± 2.36	2.11 ^A^ ± 0.88
Interaction						
PC	6.67 ^a^ ± 0.49	75.6 ^b^ ± 1.42	24.0 ^a^ ± 3.28	72.0 ^a^ ± 3.12	61.6 ^a^ ± 1.67	1.95 ^a^ ± 0.84
PCE	6.48 ^a^ ± 0.27	76.1 ^a^ ± 0.97	22.9 ^a^ ± 2.85	73.0 ^a^ ± 4.62	58.6 ^b^ ± 2.09	2.16 ^a^ ± 1.06
NC	6.41 ^A^ ± 0.24	77.22 ^A^ ± 1.27	21.6 ^A^ ± 1.95	68.4 ^B^ ± 4.31	57.9 ^B^ ± 1.41	2.15 ^A^ ± 0.96
NCE	6.42 ^A^ ± 0.24	76.59 ^A^ ± 1.20	23.4 ^A^ ± 2.84	73.1 ^A^ ± 2.54	59.9 ^A^ ± 2.60	2.06 ^A^ ± 0.80
Nutrition	0.15	0.16	0.33	0.17	0.09	0.93
Protease	0.39	0.91	0.72	0.02	0.45	0.90
Interaction	0.37	0.01	0.14	0.14	0.000	0.59
C.V. (%)	6.48	2.05	16.16	6.79	4.43	58.9

Results presented as mean ± standard deviation. Different superscript letters indicate significant differences among treatments (*p* < 0.05, ANOVA followed by Tukey’s test). In the main effects: a, b differ in relation to nutrition (PC vs. NC) and protease (0 g/t vs. 250 g/t). In the interaction: a, b (PC vs. PCE); A, B (NC vs. NCE). C.V. (%) coefficient of variation.

## Data Availability

The data presented in this study are available on request from the corresponding author.

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
