# Peer review of "Evaluation of the Effectiveness of a Matrix of Exogenous Proteases in the Nutrition of Shrimp Penaeus vannamei"

_animals, 2025, doi:10.3390/ani15101410_

Round 1
Reviewer 1 Report (Previous Reviewer 1)
Comments and Suggestions for Authors
I have carefully compared the previous and revised versions of the manuscript. Since the authors have thoroughly addressed and improved the manuscript based on my previous comments in the new revised version, I have no further comments. I recommend that the authors refer to other reviewers' suggestions for further revisions.
Author Response
We sincerely thank the reviewer for the thorough evaluation of our revised manuscript and for recognizing the improvements made in response to previous comments.
We greatly appreciate your positive recommendation and constructive feedback, which were instrumental in enhancing the quality and clarity of our work.
All the best,
Matheus Ramalho Lima

Reviewer 2 Report (Previous Reviewer 2)
Comments and Suggestions for Authors
- The conclusion did not properly summarize the work, which did not show the interest of the work nor how the work is processed.
- Figure 1, why use comma as dot? I can never accept the wrong math mark. It should be 11.07 instead of 11,07
- What is C.V %? I did not find the abbreviation explanation.
Author Response
We sincerely thank the reviewer for their constructive feedback, which helped us improve the clarity and scientific rigor of our manuscript. Please find our responses to each point below:
Reviewer Comment 1: “The conclusion did not properly summarize the work, which did not show the interest of the work nor how the work is processed.”
Response:
We appreciate this comment and have substantially revised the conclusion to clearly summarize the study's objective, methodology, key findings, and the broader relevance of the results. The revised conclusion now reads:
“This study evaluated the effects of supplementing a blend of acid and alkaline proteases at 250 g/t in diets for Penaeus vannamei. The enzyme supplementation improved nutrient utilization, which translated into better growth performance, higher yield, and enhanced meat quality. These outcomes demonstrate the effectiveness of using protease blends to optimize digestive efficiency and production metrics in shrimp farming. Given the increasing demand for sustainable and efficient aquaculture practices, the use of proteases represents a promising nutritional strategy for improving productivity in intensive shrimp culture systems.”
Reviewer Comment 2: “Figure 1, why use comma as dot? I can never accept the wrong math mark. It should be 11.07 instead of 11,07.”
Response:
Thank you for bringing this to our attention. We have corrected all decimal formats in Figure 1, replacing commas with dots (e.g., "11,07" is now "11.07") to conform to international scientific formatting standards.
Reviewer Comment 3: “What is C.V %? I did not find the abbreviation explanation.”
Response:
We apologize for the omission. "C.V %" refers to the Coefficient of Variation. We have now included this definition in the Table legends.
Additional Note:
In addition to Figure 1, we also reviewed Figure 2 and applied the same corrections to ensure consistent use of decimal points and appropriate notation throughout the manuscript.
All the best,
Matheus Ramalho de Lima
Reviewer 3 Report (Previous Reviewer 3)
Comments and Suggestions for Authors
Thank you for addressing the comments from reviewers. However, there are still points that need further clarification.
(1) The formulation of the negative control (NC). Thank you for explaining the rationale for the diets that the authors would like to test if enzyme supplement can compensate for lower availability of animal protein with fishmeal being the target ingredient. Why choose this level?The % fish meal reduction from 9.788% to 9.528% (and substitute with soybean meal and corn, based on Table 1). How was this 0.26% decrease in fishmeal come about? Any scientific or practical rationale for this level that warrant another treatment? Other studies comparing the reduction in fish meal tended to use a larger difference. Please explain your rationale.
(2) Villi height and absorption area for the PC treatment. Thank you for adding the details on histological analysis. Still, it is strange that no villi were found in the PC shrimp considering four shrimp were sampled per treatment. Do you have any explanation or speculation for this observation? This is unusual results and could affect how the results are interpreted. Please discuss in your results and discussion.
Comments on the Quality of English LanguageEnglish is okay, but still required some minor editing.
Author Response
Thank you for addressing the comments from reviewers. However, there are still points that need further clarification.
(1) The formulation of the negative control (NC).
Thank you for explaining the rationale for the diets that the authors would like to test if enzyme supplement can compensate for lower availability of animal protein with fishmeal being the target ingredient. Why choose this level? The % fish meal reduction from 9.788% to 9.528% (and substitute with soybean meal and corn, based on Table 1). How was this 0.26% decrease in fishmeal come about? Any scientific or practical rationale for this level that warrants another treatment? Other studies comparing the reduction in fish meal tended to use a larger difference. Please explain your rationale.
Response:
We appreciate the reviewer’s follow-up and the opportunity to clarify further. The 0.26% reduction in fishmeal (from 9.788% to 9.528%) was established based on the nutritional contribution (matrix value) of the protease blend used in this study. According to the manufacturer’s data, the enzyme supplementation is expected to enhance the digestibility of protein by approximately 1%, along with improved availability of energy and amino acids such as methionine, lysine, and threonine.
To reflect this added nutritional contribution, we applied a proportional reduction in fishmeal and rebalanced other ingredients (mainly soybean meal and corn) to maintain iso-nutritional profiles across treatments. Although the absolute reduction in fishmeal may appear small, it represents a deliberate and industry-relevant adjustment based on the precision formulation approach commonly used in commercial aquafeeds. Our objective was not to evaluate the impact of large-scale fishmeal replacement, but rather to validate the matrix value of the enzyme under practical formulation scenarios.
We have now expanded this explanation in the revised Materials and Methods and Discussion sections to emphasize the scientific rationale for the fishmeal reduction level.
(2) Villi height and absorption area for the PC treatment.
Thank you for adding the details on histological analysis. Still, it is strange that no villi were found in the PC shrimp considering four shrimp were sampled per treatment. Do you have any explanation or speculation for this observation? This is unusual results and could affect how the results are interpreted. Please discuss in your results and discussion.
Response:
We thank the reviewer for this important observation. Indeed, the absence of measurable villi in the PC group was unexpected. While the submucosa and muscular layers remained intact, the mucosal damage may have resulted from local inflammation, stress-induced enteropathy, or possibly minor degradation during sample fixation.
Although these changes were not observed in other groups, they prevented consistent morphometric assessment of villus height and absorption area in the PC treatment. We have now included a discussion of this issue in the Results section of the revised manuscript, clarifying that this anomaly may reflect either a transient biological response or a technical artifact.
Additionally, in the revised Materials and Methods section, we have clarified the rationale for the fishmeal reduction in the NC diet and provided further details on the enzyme supplementation strategy to highlight the relevance of this approach under practical feed formulation scenarios.
All the best,
Matheus Ramalho Lima
This manuscript is a resubmission of an earlier submission. The following is a list of the peer review reports and author responses from that submission.
Round 1
Reviewer 1 Report
Comments and Suggestions for Authors
The study presents a well-structured experimental design with a comprehensive set of evaluated parameters. The data broadly support the conclusions. However, several critical aspects require major revision before the manuscript can be considered for publication. Specifically, the introduction lacks sufficient background on the acid and alkaline proteases used in this study, and the data presentation does not meet standard publication requirements.
1. Introduction
The Introduction section does not provide an adequate review of previous studies on acid and alkaline proteases, including their application in different species and under various conditions. It is essential to discuss the effectiveness of these enzymes in prior research and clearly highlight the novelty of this study. A thorough revision is required to strengthen the rationale and significance of the research.
2. Data Presentation and Tables
Table 1: This table describes the aquaculture conditions, which is necessary for the study but does not directly impact the results and conclusions. It is recommended that this table be moved to the supplementary materials.
Table 2: The description of feed composition using (g/t) should be carefully checked to ensure it is scorrect.
Table 3, Table 4, and Table 5: The data presentation format does not conform to standard scientific reporting. It is recommended that all indices be presented as mean ± SD to facilitate a clearer understanding of the effects of PC, NC, and proteases on the measured parameters.
3. Experimental Procedures
The description of biometry and feed adjustment is unclear. The phrase “Every 15 days, the animals were captured at 20% of their initial biomass for biometry and feed adjustment” lacks details on the post-capture procedures. A more precise description should be provided regarding handling, measurement, and adjustment protocols.
4. Figures
Figure 1: Error bars are missing, which is a fundamental requirement for data visualization. Error bars should be included to indicate variability and statistical reliability.
Figure 2: The error bars appear abnormal and need correction to ensure the data representation is accurate and meaningful.
Reviewer 2 Report
Comments and Suggestions for Authors
- The work looks good, but some formats need revision. For example, in all the tables, there are some superscripts 'A' or 'a', there is no clear explanation about why those are listed.
- The hyperlink for example, Table 4 is duplicated.
- Why there are so many words underlined? Please make the format clear for publication style.
- Fig3, why there is only one bar in the left. Also, the superscripts are weird.
- Table 1, Trat1, why the temperature has so large variance compared with others?
Reviewer 3 Report
Comments and Suggestions for Authors
The manuscript examined the effect of exogenous protease supplement on growth performance, product quality and intestinal morphology. The experiment was well designed with sufficient replicate and appropriate statistical analyses. Results presentation needs some work, but the interpretation is good. The manuscript needs some work with the use of English. While the manuscript can be understood, there is some issues with word choices. Many unconventional words and units were used that can affect the reader’s understanding of the manuscript.
(1) Line 113: “Negative Control (NC), which presented a nutritional reduction based on 100% of the blend of acid and alkaline proteases (energy, crude protein, methionine + cystine, methionine, lysine and threonine)." Please explain the rationale for the nutritional reduction diet in a different way. What was the target reduction? How was the target chosen?
(2) Line 117-118: Please double check the concentration of protease added to the diet. I’m not quite familiar with “g/t” unit as most paper generally used % or g/kg. Is 250 g/t mean 250 gram per tonne? Equivalent to 250 mg/ kg? Please consider convert the unit to more familiar ones.
(3) Table 2.
- The unit “g/t” does not make sense here. With a combined total of 100, the unit should be as %.
- Soybean meal 45%, Corn 8%, Fishmeal 53% What was this number referring to? Crude protein? Please specify clearly in the table.
- There are some unusual ingredients used for grow-out shrimp feed formulation. Artemia biomass, seaweed, lactic acid, sodium lactate. What was the reasoning?
- The protease blend concentration here is 0.025 (g/t) only. Please double your number and unit. So is this in percent? What about 250 g/t mentioned in the text (Line 117-118)
(4) Please include the proximate analysis results of the feed. This information is crucial for a study involving feed and feed additive to show that the pelleted feed from the formulation did deliver the nutrient composition as planned. It is not sufficient to show only the calculated composition. Without this information, the results and conclusions cannot be valid.
(5) Line 133: LIVE (%). Please consider “Survival rate (%)”
(6) Line 141: “head and carapace“ Please double check. The header 2.4 referred to “headless and shell-less shrimp”. Carapace refers to only the exoskeleton of the upper section (head) of the shrimp, while shell refers the exoskeleton of the entire body. What was actually measured here?
(7) Line 149 “humidity” Please consider using “moisture” (Also, at other points through out the manuscript. Table 4 and in the text)
(8) 2.7 Histological analysis: How many repeated measurement was made for each shrimp? (how many section of intestinal wall was measured for thickness? How many villi were measured for length?)
(9) All results tables.
- Please consider a different way to present this results. The current version is difficult to understand. Please make sure to distinguish between results from the experiment and statistical analysis results (P-value, main effects/ interactions). Not all of these needs to be presented at once. You may consider putting the details results as supplements.
- The title of the table needs to describe all the information presented. (including explain your statistical analysis results)
- Include units in your table.
- Show your standard variation or standard error terms in the table (Table 1 had it, but not in other results table).
(10) Line 213 “shrimp fillet” The term “fillet” is more commonly associated with fish. Please continue using “headless and shell-less” shrimp as in Line 206 to prevent confusion.
(11) Fig 1. Please include error bars in your figure. Also, keep the treatment acronym consistent. PC, PCE, NC, NCE.
(12) Fig 3. Why was there no measurement for some treatment? (No AV and AA for PC treatment?) Are these missing results? If so, I do not think this results is appropriate to include in the manuscript and please consider removing the section. Also, keep the treatment acronym consistent. PC, PCE, NC, NCE.
(13) Line264 “fattening phase” - do you mean “grow-out”?
(14) Discussion.
- Line 282-291. Please provide specific examples or concrete results when compare your results with previous publication. How much protease was used in the other experiment? Fishmeal concentration? How were these concentrations compared to your study?
- Line 294: “shrimp fillet” Not sure if this is a regional use of the word, but “headless and shell-less” is more common for shrimp.
Please consider using more conventional terminology similar to other papers in the field that most reader would easily understand. Keep the terms and acronyms consistent throughout.